# Microstructural Evolution and Mechanical Properties of Extruded AZ80 Magnesium Alloy during Room Temperature Multidirectional Forging Based on Twin Deformation Mode

**DOI:** 10.3390/ma17205055

**Published:** 2024-10-16

**Authors:** Rou Wang, Fafa Yan, Jiaqi Sun, Wenfang Xing, Shuchang Li

**Affiliations:** 1Ningbo Surface Engineering Research Institute Co., Ltd., Ningbo 315177, China; zjbmzx@126.com; 2Ningbo Branch of Chinese Academy of Ordnance Science, Ningbo 315103, China; yanfafa_88@163.com (F.Y.); jiaqisun@zju.edu.cn (J.S.); 13805869652@163.com (W.X.); 3School of Aerospace Engineering, North University of China, Taiyuan 030051, China

**Keywords:** magnesium alloy, multidirectional forging, twinning, mechanical properties

## Abstract

This study investigates the preparation of ultrahigh-strength AZ80 magnesium alloy bulks using room temperature multidirectional forging (MDF) at different strain rates. The focus is on elucidating the effects of multidirectional loading and strain rates on grain refinement and the subsequent impact on the mechanical properties of the AZ80 alloy. Unlike hot deformation, the alloy subjected to room temperature MDF exhibits a lamellar twinned structure with multi-scale interactions. The key to achieving effective room temperature MDF of the alloy lies in combining multidirectional loading with small forging strains per pass (6%). This approach not only maximizes the activation of twinning to accommodate deformation but ensures sufficient grain refinement. Microstructural analysis reveals that the evolution of the grain structure in the alloy during deformation results from the competition between {10$\bar{1}$2} twinning or twinning variant interactions and detwinning. Increasing the forging rate effectively activates more twin variants, and additional deformation passes significantly enhance twin interaction levels and dislocation density. Furthermore, at a higher strain rate, more pronounced dislocation accumulation facilitates the transformation of twin structures into high-angle grain boundaries, promoting texture dispersion and suppressing detwinning. The primary strengthening mechanisms in room temperature MDF samples are grain refinement and dislocation strengthening. While increased dislocation density raises yield strength, it reduces post-yield work hardening capacity. After two passes of MDF at a higher strain rate, the alloy achieves an optimal balance of strength and ductility, with a tensile strength of 462 MPa and an elongation of 5.1%, significantly outperforming hot-deformed magnesium alloys.

## 1. Introduction

Magnesium (Mg) alloys, known for having the lowest density among currently applied metal materials, not only exhibit high specific strength but also possess excellent electromagnetic shielding, electrical conductivity, and thermal conductivity properties [1,2]. These characteristics make them highly promising for applications in fields such as transportation and aerospace. However, despite their numerous advantages, the relatively low absolute strength of Mg alloys has limited their application in most load-bearing structural components [3].

Existing research indicates that two of the most effective methods for enhancing the performance of Mg alloys are refining their microstructure through thermomechanically coupled severe plastic deformation (SPD) and adding rare earth (RE) elements to increase strength and the age-hardening response [4]. For instance, Wang et al. [5] reported that the yield strength of a bimodal-structured Mg–RE alloy prepared by hot extrusion followed by rapid cooling reached 470 MPa. Although the addition of RE elements can improve both room and high-temperature properties of Mg alloys, the high cost restricts their widespread future application. Alternatively, SPD techniques such as equal channel angular pressing (ECAP) and repetitive upsetting-extrusion (RUE) could effectively refine the grain size of Mg alloys, reduce deformation texture, and significantly improve performance [6,7]. For example, the grain size of an AZ31B Mg alloy can be refined to ~2.5 μm after a total of six passes of ECAP at 200–250 °C [8]. However, SPD typically involved multiple passes, reducing processing efficiency and increasing deformation costs.

Due to the poor room temperature ductility of Mg alloys, processing is usually conducted at higher temperatures. However, high deformation temperatures lead to significant dynamic recovery during plastic deformation, greatly weakening the effects of dislocation strengthening. Consequently, it is challenging for Mg alloys to achieve cold deformation-strengthening effects similar to those of aluminum (Al) alloys and steels. Recent studies have found that activating tensile twinning at room temperature can enable large plastic deformation and simultaneously strengthen Mg alloys. For instance, Wang et al. [9] introduced a substantial number of tensile twins and dislocation structures in a rolled AZ80 Mg sheet by applying 6% cold deformation. They observed that the introduction of dislocations accelerated the age-hardening response, significantly enhancing the sheet’s performance. Through multidirectional forging (MDF) at room temperature, Miura et al. [10] increased the tensile yield strength (TYS) of an extruded AZ80 Mg alloy to 500 MPa, revealing that nanoscale twinning and dislocation strengthening were the main reasons for the performance improvement. However, these results were reported in terms of true stress–strain, which may involve certain conversion errors. Zhang et al. [11] prepared AZ80 Mg alloy blocks with an ultimate tensile strength (UTS) of 500 MPa and over 20% elongation (EL) through 30 cycles of room-temperature MDF. However, the TYS was insufficient, being only 216 MPa in terms of true stress. Yan et al. [12] proposed a MDF method with low–high strain alternations (3% and 6%) to produce nanoscale twinned AZ80 Mg alloys. After four passes of such MDF, the grain size was refined to ~300 nm, and combined with aging treatment, the alloy achieved a true TYS exceeding 300 MPa with a room temperature ductility of 14.6%.

The cold deformation strengthening mechanism of Mg alloys mainly stems from grain refinement caused by twinning and work hardening induced by dislocations [13]. However, there are few studies on the microstructural evolution characteristics during the room temperature MDF of Mg alloys, especially the grain refinement features. Additionally, the effects of different strain rates on the microstructure and properties of room temperature MDF samples have not been reported. It is well known that the introduction of twins in Mg occurs under strong texture conditions. Therefore, this study conducted room temperature MDF on a commercial extruded AZ80 Mg alloy at two different strain rates. This study aims to reveal the twinning refinement characteristics under different strain amounts and rates and their effects on the microstructure and mechanical properties of the AZ80 alloy. The research findings provide valuable insights for advancing the application of room temperature MDF in the development of high-performance Mg alloys.

## 2. Materials and Methods

The material used in this experiment was an extruded AZ80 Mg alloy (Mg-8.0Al-0.5Zn-0.11Mn, wt.%) rod, provided in the dimensions of 120 mm in diameter and 1000 mm in length. Samples with dimensions of 32 mm (ED) × 31 mm (TD) × 30 mm (RD) were cut from the rod along the cross section using wire electrical discharge machining. These samples underwent solution treatment (ST) at 420 °C for 8 h to eliminate precipitated phases and achieve a more uniform grain structure. Figure 1 shows the schematic diagram of the MDF process. Considering the basal fiber texture characteristics of the extruded bar (Figure 2), the MDF followed the ED–TD–RD deformation path to rapidly introduce tensile twins. The designed forging passes for this experiment were 1 to 3, with a strain of 6% on each face, corresponding to total strains of 0.18, 0.36, and 0.54, respectively. To investigate the effect of the forging rate on deformation behaviors, two different forging rates were employed, 0.006 s^−1^ (Group 1) and 0.06 s^−1^ (Group 2). These rates were chosen to reflect practical conditions, where the forging rate cannot be excessively high. Due to the small forging strain per pass, free forging was adopted. The MDF was performed on a 100-ton servo-hydraulic press (Shandong Dingrun Forging Co., Ltd., Zaozhuang, China).

The microstructure of the deformed samples was observed using electron backscatter diffraction (EBSD), with samples taken from the middle of the ED–TD plane. The samples were initially ground with SiC paper up to 7000 grit, then mechanically polished and finally, the residual stress layer was removed using a Leica EMS-102 ion thinning instrument (Leica Corporation, Wetzlar, Germany) with an operating voltage of 6.0 kV, a current of 2.3 mA, and a duration of 30 minutes. EBSD was conducted on an SU5000 scanning electron microscope (SEM; Hitachi Inc., Tokyo, Japan) equipped with an EDAX-TSL (EDAX Inc., Mahwah, NJ, USA) system, with a scanning step size of 200–400 nm. Data on grain size, grain boundary characteristics, and texture were obtained from EBSD and analyzed using the orientation imaging microscopy (OIM) analysis-v7.3 software (EDAX Inc., Mahwah, NJ, USA). In the OIM analysis software, the grain size was calculated as the equivalent diameter. When the grain boundary misorientation was greater than 15 °, it is defined as a high-angle grain boundary (HAGB). In order to obtain more accurate micro-texture information, samples with each deformation parameter were tested 2–3 times, and the texture statistical results were obtained by combining 2–3 EBSD data with the same scanning step size. The texture intensity was calculated through the multiplication of random distribution (MRD). A SmartLab-3KW X-ray diffraction instrument (XRD; Riken Electric Co., Ltd., Tokyo, Japan) was used to analyze the dislocation density in the deformed samples, employing Cu-Kα radiation (wavelength λ = 0.154 nm) at 45 kV and a tube current of 200 mA. The XRD measurements were performed over a 2θ range from 30° to 40° using a scanning step of 0.02° and a scanning speed of 0.6°/min. Tensile specimens with gauge dimensions of 8 mm × 2 mm × 2 mm were taken from the central region of the samples. Tensile tests were conducted on an Instron 3382 electronic universal testing machine (Instron Inc., Canton, MA, USA) at an initial strain rate of 1 × 10^−3^ s^−1^, with each parameter tested at least three times.

## 3. Results

### 3.1. Initial Microstructure

Figure 2a display the inverse pole figure (IPF) coloring map of the extruded AZ80 Mg alloy rod after solution treatment at 420 °C for 8 h. The ST sample exhibits a relatively uniform equiaxed grain structure with an average grain size (AGS) of ~60 μm. The XRD results in Figure 2c confirm that only the diffraction peaks of Mg were observed after ST, indicating that the precipitated phases have been completely dissolved into the matrix. The texture of the ST sample shows a typical extrusion texture, with the basal pole parallel to the ED and a uniform distribution of [10-10]-[11-20] double fiber components (Figure 2b). It is well known that {10$\bar{1}$2} tensile twinning in Mg is significantly activated when the c-axis of the parent grain is parallel to the compressive stress direction. Therefore, the ED–TD–RD deformation path was chosen for MDF to rapidly introduce tensile twins.

### 3.2. Microstructure of MDF Samples

Figure 3 shows the IPF coloring maps and corresponding grain size distribution maps for samples processed at room temperature MDF with different parameters. At room temperature deformation, thermally activated processes involving dislocation rearrangement are significantly suppressed, leading to greatly reduced dynamic recovery and dynamic recrystallization (DRX) capabilities of the metal. Therefore, the activation of tensile twinning may become the primary deformation mechanism, refining the grain structure. Post-MDF deformation, the initial equiaxed grain structure was largely replaced by a twin structure with multi-scale interactions, which closely resemble {10$\bar{1}$2} tensile twins. At the same strain rate, the density of twins increased significantly with the number of forging passes, and the grain size was notably refined. At the same number of forging passes, samples subjected to a higher strain rate exhibited higher twin density, indicating that an increased strain rate favors grain refinement. The grain refinement effect was more pronounced during the first two passes and diminished during the third pass, possibly suggesting that the stress required for twin nucleation increased with the grain refinement [14]. Compared to larger grain sizes in the initial state (AGS of ~60 μm), the AGS for the samples processed at a low strain rate (Group 1) decreased to 22.8 μm for one pass, 21.1 μm for two passes, and 16.2 μm for three passes. For Group 2, the AGS decreased to 20.6 μm for one pass, 17.6 μm for two passes, and 15.1 μm for three passes. Despite the grain size differences narrowing after three passes, the grain size distribution within 0–10 μm is higher in Group 2 with a higher strain rate, indicating more thorough refinement.

Figure 4 shows the grain boundary (GB) characteristics and the corresponding misorientation angle distribution (θ > 15°) of the MDF samples at room temperature. After MDF, the GB characteristics exhibit three distinct preferential distributions, with major peaks at ~45°, ~60°, and ~86°. The inserted GB rotation axis distribution indicates that the 45° peak corresponds to enrichment of the <20$\bar{2}$1> axis, the 60° peak to the <10$\bar{1}$0> axis enrichment, and the 86° peak to the <11$\bar{2}$0> axis enrichment. It is well known that the activation of {10$\bar{1}$2} tensile twins in Mg resulted in an 86° <11$\bar{2}$0> orientation relationship between the matrix and the twinned regions, while interactions between tensile twin variants contributed to the formation of the 60° <10$\bar{1}$0> GB [15]. This clearly indicates that twinning and twin variant activation were the dominant deformation mechanisms during room temperature MDF. Additionally, a new 45° <20$\bar{2}$1> orientation relationship also appeared frequently, which is reported to be associated with the interaction of tensile twins [16], as discussed in the next section. Comparing MDF samples with different parameters, it is evident that tensile twin boundaries (TBs) dominate the GB proportions. The relative proportion of twin variants increased with the number of forging passes, and Group 2 shows more activated tensile twin variants. Furthermore, the proportion of 45° <20$\bar{2}$1> GB increased with the number of forging passes, and in Group 2 after three passes, the 45° and 60° peaks gradually merged, forming a broader preferential peak.

Figure 5 shows the (0001) pole figures (PFs) and IPFs of MDF samples with different parameters. Under multidirectional stress, the texture of MDF samples shows significant changes compared to the initial state. The initial fiber texture became more dispersed, with the basal plane orientation deviating up to 30° from the ED, and a new ED component appeared. In Group 1 with 1–2 passes, the ED component is stronger than in other samples. The texture intensity of Group 1 is slightly weaker than that of Group 2 at the same forging pass. Clearly, the deformation characteristics of MDF involve interactive stresses in three directions. The emergence of the ED component was closely related to twin nucleation induced by loading in the ED. Meanwhile, the loading in the TD and ND during a deformation cycle could lead to the consumption of ED components through secondary twinning or detwinning. The current results suggest that such consumption was more evident at a higher strain rate, likely due to more thorough twin activation. Overall, the texture of all samples becomes significantly dispersed, potentially weakening the anisotropy caused by basal fiber texture in an extruded state.

Figure 6 presents the Kernel average misorientation (KAM) maps and corresponding distribution histograms for MDF samples with different parameters. The KAM can be used to explain the local geometrically necessary dislocations (GNDs) of crystal materials, especially for the strain distribution at GBs and phase boundaries of crystal materials after deformation. A large number of GNDs are activated during room temperature MDF deformation and continue to grow with the number of forging passes. In the one-pass samples, higher dislocation density was clearly observed in regions with high twin density and twin interactions. Notably, with the same number of passes, the dislocation density increased with the strain rate. As the number of forging passes increased to three, the difference in dislocation density between the two groups became similar. This phenomenon might be due to the similar regions of dislocation density in both samples, but it could also be influenced by the chosen step size for the EBSD test.

It is well known that reducing the step size in KAM calculations can enhance the accuracy of dislocation density estimation, but this is also influenced by time costs. Therefore, to further analyze the macroscopic dislocation density of the deformed samples, XRD was used to measure the changes in lattice parameters of three typical peaks of Mg for MDF samples under different parameters, as shown in Figure 7. During plastic deformation, changes in grain size and dislocation multiplication can alter the interplanar spacing, leading to variations in the diffraction peak width and position [17]. It is evident that the intense dislocation multiplication in cold deformation caused lattice distortion, resulting in fluctuations and broadening of the diffraction peaks. The dislocation density of crystals can be calculated using the Williamson–Hall (W–H) method [18]. The main formulas are as follows:(1)$$\delta_{hkl}=\delta_{e,hkl}+\delta_{D,hkl}$$

$\delta_{hkl}$, $\delta_{e,hkl}$, and $\delta_{D,hkl}$ represent the total full width at half maximum (FWHM), and the FWHM caused by defects and grain refinement is
(2)$$\frac{\delta_{hkl}cos\theta_{hkl}}{\lambda }=\frac{1}{D}+\frac{2esin\theta_{hkl}}{\lambda }$$

$\theta_{hkl}$ represents the corresponding peak angle of diffraction at the crystal plane; λ represents the wavelength of the electron beam emitted during XRD (0.179026 nm); *D* represents the grain size of the material; and *e* is the average micro strain. The dislocation density of the sample according to micro strain can be calculated as follows:(3)$$\rho =\frac{14.4e^{2}}{b^{2}}$$
where *b* is the Burgers vector of the Mg alloy with a value of 3.2 × 10^−4^ m.

In this study, the FWHM and the diffraction peak angles of the three main grain planes ((10$\bar{1}$0), (0002), and (10$\bar{1}$1)) for each sample were measured, and the calculated results are shown in Table 1. Consistent with the KAM calculation results, a significantly high dislocation density was introduced into the matrix during room temperature MDF, with Group 2 showing a higher dislocation density than Group 1. Typically, in thermoplastic deformation, the matrix dislocation density of magnesium is lower than 10^15^/m^2^. Unlike the KAM results, the dislocation density of the samples at three passes also shows significant differences, with the sample at a low strain rate having a dislocation density of 32.5 × 10^14^/m^2^, and the latter significantly increasing to 40.2 × 10^14^/m^2^. This indicates some differences between the GNDs calculated by EBSD and the lattice dislocations measured by XRD. Moreover, KAM calculations are influenced by factors such as selected area and sample surface quality, making them unsuitable for overall dislocation density measurements but more advantageous for a semi-quantitative characterization of local dislocation distribution. In summary, the current test results show that the dislocation density in Group 2 was higher than in Group 1.

### 3.3. Grain Refinement Behavior during Room Temperature MDF

Unlike deformation at higher temperatures, dynamic recovery and dynamic recrystallization are suppressed during room temperature MDF, so dislocation pile-up is mainly coordinated through the activation of tensile twinning. To elucidate the grain refinement behavior and its relationship with texture and dislocation density evolution in samples subjected to room temperature MDF, several representative regions (A1–A4) from Figure 3 were selected for more detailed analysis, as shown in Figure 8.

Figure 8a–d show an IPF map of a grain (A1) from sample 1P-1, along with its grain boundaries, KAM map, and (0001) PF. Based on the initial texture characteristics and GB identification in Figure 8b, it can be concluded that M is the matrix, with only one type of twin (T) appearing in the matrix. The KAM map reveals a large number of dislocations in the matrix, with these dislocations accumulating in certain regions to form LAGB bands. The (0001) PF shows that the orientation of the twin T is close to the ED direction, corresponding to the first forging in each cycle (i.e., ED loading). Since TD and ND are orthogonal to ED, the T is likely to be consumed by detwinning or further twinning during the second and third forgings of each cycle. The results indicate that detwinning is the primary deformation mechanism in this grain, simultaneously generating many dislocations in the matrix. Generally, the movement of TBs involves the cross-slip and climb of TB dislocations [19]. Continuous interactions between TBs and the matrix can promote dislocation accumulation within the twins, making twinning activation a mechanism for deformation coordination, as well as promoting grain subdivision and dislocation storage. From Figure 8a–c, it can be noted that high dislocation densities tend to accumulate at LAGBs, with some LAGBs presenting as banded structures, likely related to interactions between migrating TBs and matrix dislocations, confirming that TB motion enhanced matrix dislocation density.

Figure 8e–h show another typical grain (A2) from sample 1P-1, illustrating the interaction between twins and dislocations. More complex twin activation occurred in the matrix grain, promoting grain refinement. Based on the initial texture and forging stress states, it can be inferred that the matrix M transformed into twin variants (T_Vs_) of T_V1_ and T_V2_ under ED and TD stresses, respectively. T_V1_ further underwent secondary twinning under TD and ND stresses, forming two new twin variants (T_V1-V1_ and T_V1-V3_), promoting grain refinement. It should be noted that one tertiary twin (T_V1-V1-T_) was also observed within T_V1-V1_. The emergence of T_V2_ and T_V1-V1-T_ dispersed the initial fiber texture, weakening the ED component under ED stress. Thus, twin variant interactions partially weaken the texture, showing a regular weakening pattern rather than random orientation nucleation from DRX. GB analysis reveals not only twin and twin variant boundaries but also a 45° <20$\bar{2}\bar{1}$> GB formed within the matrix, reflecting the misorientation distribution in Figure 4. It is obvious that this boundary formed from interactions between secondary twins within twin variants, such as T_V2_ and T_V1-V1_. The KAM distribution shows that as grain refinement progresses, matrix dislocation density further increased, with higher KAM values formed in regions of twin interaction and fine grains (Figure 8g, black elliptical region), indicating that twin variant interactions are more effective at trapping matrix dislocations.

It can be inferred that at lower deformation passes, single twin activation was the primary deformation mode, with twin expansion unaffected by other twin variants, leading to detwinning behavior. This stage primarily promoted dislocation accumulation within the matrix and at GBs. As overall dislocation density increased, dislocation movement resistance rose significantly, reducing slip activation capacity. Single twin activation and slip activation could no longer fully coordinate deformation, ultimately activating different twin variants within the matrix. On the other hand, the interaction between twin variants contributes to grain subdivision and refinement, which in turn could weaken detwinning behavior through enhancing the Hall–Petch effect. Additionally, it also promoted dislocation pile-up, further stimulating the nucleation of more twins. Again, this process accelerated grain refinement and texture dispersion and increased the dislocation density in the matrix.

Figure 8i–l and Figure 8m–p illustrate typical twin evolution behaviors from samples 2P-1 (A3) and 3P-2 (A4), respectively. As deformation passes increased, more fine grains appeared, primarily due to more extensive twin refinement. Matrix dislocation density also significantly increased compared to earlier passes. GB distribution shows that some GB misorientations of fine twinned grains deviated from typical twin and twin variant boundaries by over 5°, suggesting that continuous dislocation accumulation has disrupted the coherence of TBs and twin variant boundaries, pushing them towards dynamic recrystallization-like grains. These results are consistent with the misorientation distribution in Figure 4. Moreover, it is evident that grain orientation evolution remained consistent with earlier process passes, but more twin variant activation promoted more significant grain subdivision and relative orientation dispersion, reducing overall texture intensity.

The local microstructural evolution results show that grain refinement in room temperature MDF is due to the activation of tensile twins and their variants. TB migration and preferential dislocation accumulation within twins significantly increase matrix dislocation density. Matrix dislocation density also greatly affected subsequent twin behavior, with higher dislocation densities promoting more twin variant interactions, further increasing dislocation density. The analysis indicates that samples with high strain rates show higher proportions of tensile twin variant activation, significantly promoting grain refinement and overall dislocation density improvement. It can be inferred that the activation of a high proportion of twin variants under a higher strain rate during MDF is likely related to the inability of dislocation slip to sufficiently accommodate deformation. As we know, achieving more coordinated plastic deformation in Mg alloys requires the activation of pyramidal <c + a> slip ({11$\bar{2}$2} <11$\bar{2}\bar{3}$>) [20]. However, at room temperature, <c + a> slip retains a very high critical resolved shear stress (CRSS), making it difficult to activate extensively. As a result, tensile twinning, which has a lower CRSS, can effectively contribute to plastic deformation, even when the matrix is in orientations unfavorable for twinning activation [21].

### 3.4. Mechanical Properties

Figure 9 shows the tensile properties of MDF samples tested along the ED, with the corresponding data summarized in Table 2. It is evident that the tensile properties of all samples were significantly improved after MDF deformation, with some achieving a UTS up to 500 MPa. Notably, after the yield stage, all tensile curves exhibit normal strain hardening behavior, indicating that dislocation slip was the primary deformation mechanism during deformation. This suggests that TBs can also act as obstacles to dislocation motion, rather than merely being high migrating. It can be seen that with an increase in the number of forging passes, the tensile strengths of all samples gradually improved, while their ductility decreased. At lower passes, samples from Group 1 exhibited better ductility, with the sample 1P-1 showing the highest elongation of 6.5% but the lowest UTS of 420 MPa. However, with an increase in the number of forging passes, the ductility loss in Group 1 was greater than in Group 2. It should be noted that the samples 2P at both low and high strain rates achieved a good balance of strength and moderate ductility.

Figure 10 compares the mechanical properties of samples prepared in this study with some typical thermomechanically processed high Al content AZ series alloys. Traditionally, the tensile strength of AZ80 or AZ91 alloys rarely exceeded 400 MPa under conventional thermomechanical processing. However, after room temperature MDF, the tensile strength of AZ80 alloys in this study all exceeded 400 MPa. It is known that grain structure, texture, dislocation density, and possible precipitate phases are key factors affecting the properties of Mg alloys. In traditional thermomechanical processing, dynamic grain refinement induced by plastic deformation, texture strengthening formed under directional stress, and subsequent appropriate heat treatment are crucial for improving the properties of Mg alloys [4]. In contrast, during room temperature MDF based on twinning-coordinated deformation, grain refinement is not as significant, falling short of the grain refinement levels reported for thermomechanical processing. For instance, Kang et al. [22] reported that the AGS of an AZ91 Mg alloy could be refined to ~1 μm after extrusion at 250 °C. However, dislocation structures are maximally retained during room temperature deformation.

It is well known that Mg alloys exhibited poor plasticity at room temperature, primarily relying on basal <a> slip ({0001} <11$\bar{2}$0>) and {10$\bar{1}$2} tensile twinning for plastic deformation. Typically, the activation and rapid expansion of tensile twins could cause severe stress concentration and eventual fracture in the alloy [27]. On the one hand, the nucleation of tensile twins drove the matrix towards hard orientations, weakening the coordination of deformation; on the other hand, the migration of TBs inherently involved dislocation movement and promoted dislocation accumulation within twins, both of which reduced the plasticity of the alloy [19]. Recent studies [28,29,30] also show that high dislocations within tensile twins were composed of basal I-type stacking faults and <c + a> dislocations, resulting from interactions between TBs and the matrix’s <a> dislocations during TB expansion. The key to room temperature MDF is the small strain amplitude in each pass (only 6%), which effectively activated tensile twins without causing excessive work hardening. Alternating stress states favor multi-scale twin activation without causing alloy cracking. This process ultimately achieved significant dislocation storage and a certain degree of grain refinement.

It should be considered that the main strengthening mechanisms in room temperature MDF samples are twinning refinement and dislocation strengthening. Increasing the strain rate induced more twins and twin variant activation, suppressing detwinning, and significantly enhancing alloy strength. However, excessively high dislocation densities also severely reduced work hardening capacity beyond the yield stage, deteriorating plasticity. Alloys after 2P of MDF exhibited excellent strength with some room temperature plasticity. Maintaining high strength with adequate plasticity indicates that twin-structured Mg alloys are more advantageous than conventional grain boundary-structured Mg alloys. It is well known that the GB structure significantly impacts alloy plasticity. For conventional thermomechanically processed Mg alloys, grain refinement greatly enhanced intergranular deformation coordination, activating non-basal slips and promoting uniform deformation [31]. However, for twin-structured Mg alloys, there is debate about whether TBs serve the same function as GBs. On the one hand, TBs were revealed to act as obstacles to dislocation movement, thus strengthening the alloy; on the other hand, interactions between TBs and dislocations could promote dislocation transmission and alleviate boundary stress concentration. Additionally, under favorable loading conditions, TBs also exhibited high migration capability. Kuang et al. [32] suggested that in a pre-twin-structured AZ31 Mg alloy, prismatic <a> slip ({10-10} <11$\bar{2}$0>) and <c + a> slip were significantly activated in the matrix, enhancing alloy plasticity. Unlike the macroscopic Hall–Petch effect, research indicated that TBs have a lower micro-Hall–Petch slope, reducing the ratio of non-basal slip to basal <a> slip. Another study led by Chen et al. [33] also suggested that AZ31 Mg alloys with pre-twin structures improved both strength and toughness, attributed to enhanced basal <a>-prismatic <a> or <c + a> slip transfer efficiency and more uniform deformation promoted by twin interfaces. Although the impact of TBs on plasticity requires further research, they significantly contribute to alloy plasticity.

Additionally, texture is another key factor affecting the plasticity of Mg alloys [34]. Due to multidirectional loading, the strong texture state of the material was disrupted, weakening the strong basal fiber texture in the extruded state. As shown in Figure 11, the Schmid factor (SF) distributions for basal <a> slip during ED loading of MDF and ST AZ80 alloys show that the texture weakening facilitated basal <a> slip. As the number of deformation passes increased, the SF for basal <a> slip gradually increased, with Group 2 having higher average SF values than Group 1. On the one hand, as the number of deformation passes increased, the grain refinement caused by twin and twin variant subdivisions became significant, weakening single texture type and promoting relative grain orientation dispersion. On the other hand, under higher deformation passes and strain rates, TBs could deviate more from the fixed twin orientation, promoting grain orientation dispersion. The activation of basal <a> slip weakened the dominance of single prismatic <a> slip activation in the extruded state, facilitating the complementary deformation mechanism and promoting multi-slip coordinated deformation, potentially enhancing alloy plasticity.

## 4. Conclusions

This study used room temperature MDF to prepare an ultrahigh-strength AZ80 Mg alloy with a twinned lamellar structure under different strain rates (0.006 s^−1^ and 0.06 s^−1^). The research elucidated the grain refinement behavior of the alloy under multidirectional loading and the effects of strain rate on grain refinement and mechanical properties. The main conclusions are as follows:The AZ80 alloy subjected to room temperature MDF exhibited a multi-scale grain structure dominated by tensile twins and their variants. With increasing deformation passes (1P–3P), the twin density increased, and the higher strain rate resulted in better grain refinement compared to the lower strain rate. After three passes of MDF at a high strain rate, the grain size of the alloy was greatly refined from the initial 60 μm to 15.1 μm.The interaction between tensile twins and twin variants during room temperature MDF promoted grain refinement, while detwinning caused grain coarsening. At lower deformation passes, detwinning was significantly promoted. However, with increasing deformation passes, the concurrent increase in the number of twin variants and dislocation density raised the stress required for detwinning and induced a new cycle of twin nucleation for coordinated deformation, restraining grain coarsening.The alloy showed a higher proportion of twin interactions and dislocation accumulation at higher strain rates. Moreover, under high strain rates and higher deformation passes, the twin structure gradually deviated from the ideal misorientation and evolved into new HAGBs, promoting texture dispersion and suppressing detwinning.Tensile tests indicated that all MDF samples exhibited tensile strengths exceeding 400 MPa, significantly superior to those in the hot-deformed state. As deformation passes increased, the rise in dislocation density enhanced the yield strength but reduced the work hardening capacity beyond the yield point. After MDF of 2P at a higher strain rate, the alloy achieved a good balance of strength and ductility, with a UTS of 462 MPa and an elongation of 5.1%.The main strengthening mechanisms for room temperature MDF AZ80 alloys were grain refinement and dislocation strengthening. As a special type of GB, TBs can not only enhance alloy strength through the Hall–Petch effect, but also effectively enhance the transmission of non-basal slip, avoiding excessive stress concentration. This is the reason why the multi-scale twinned Mg alloy has high strength and appropriate plasticity.

## Figures and Tables

**Figure 1 materials-17-05055-f001:**
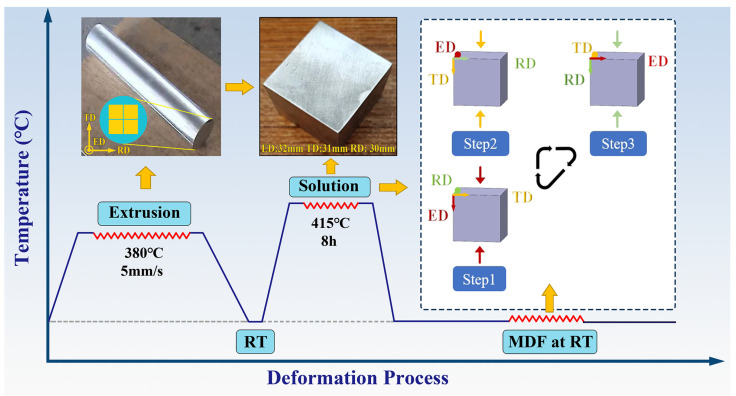
Schematic diagram of the MDF process (where ED represents the extrusion direction of the extruded bar, while TD and RD represent the transverse directions).

**Figure 2 materials-17-05055-f002:**
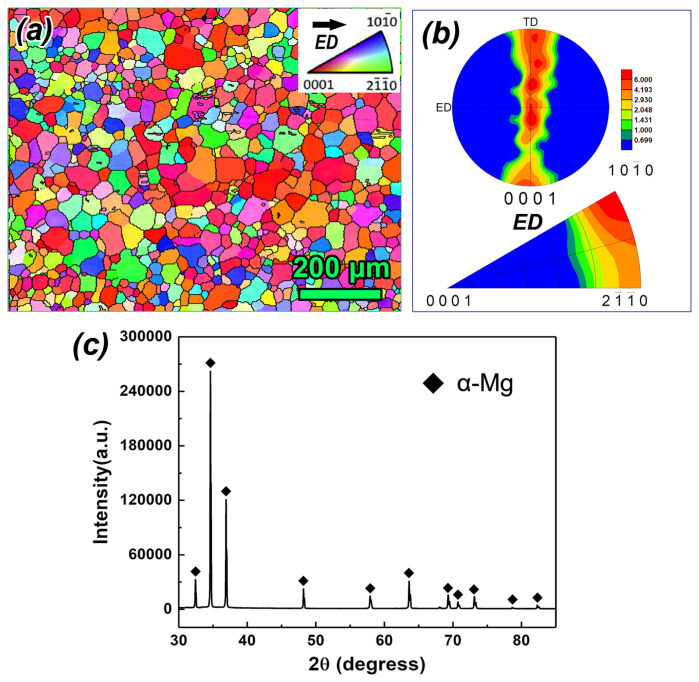
(**a**) The IPF coloring map, (**b**) texture, and (**c**) XRD pattern of as-extruded AZ80 Mg alloy after ST.

**Figure 3 materials-17-05055-f003:**
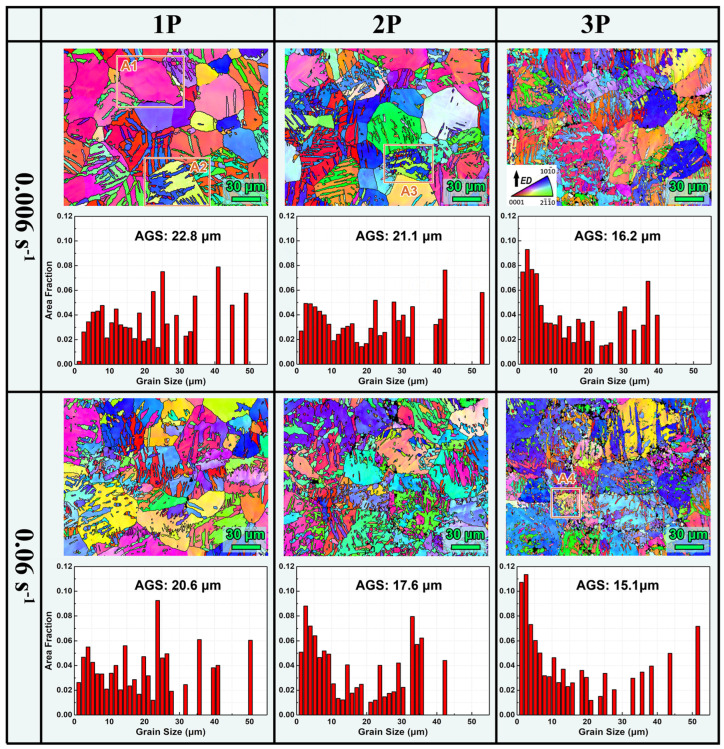
Microstructures and grain size distributions of AZ80 Mg alloy processed at room temperature MDF with different parameters.

**Figure 4 materials-17-05055-f004:**
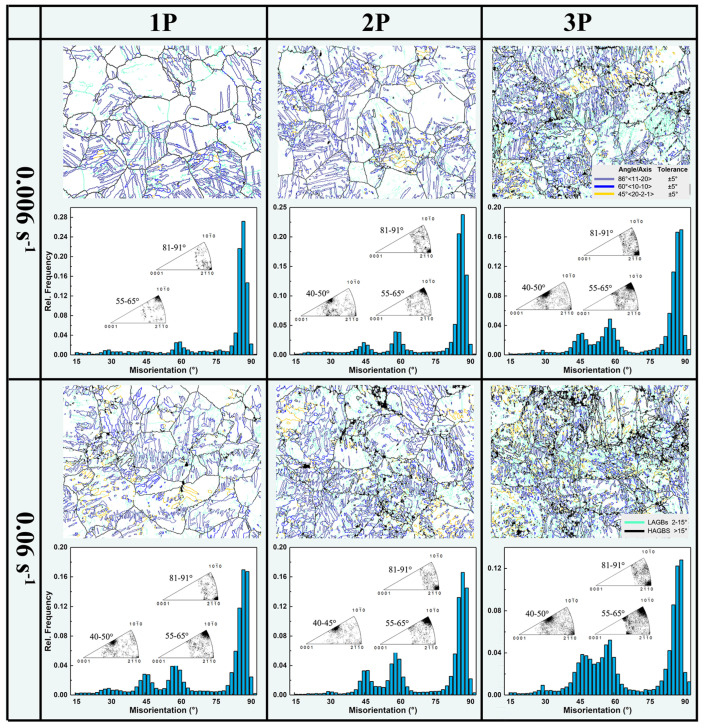
GB characteristics are highlighted with inserted GB rotation axis distribution and corresponding misorientation angle distributions of AZ80 Mg alloy processed at room temperature MDF with different parameters.

**Figure 5 materials-17-05055-f005:**
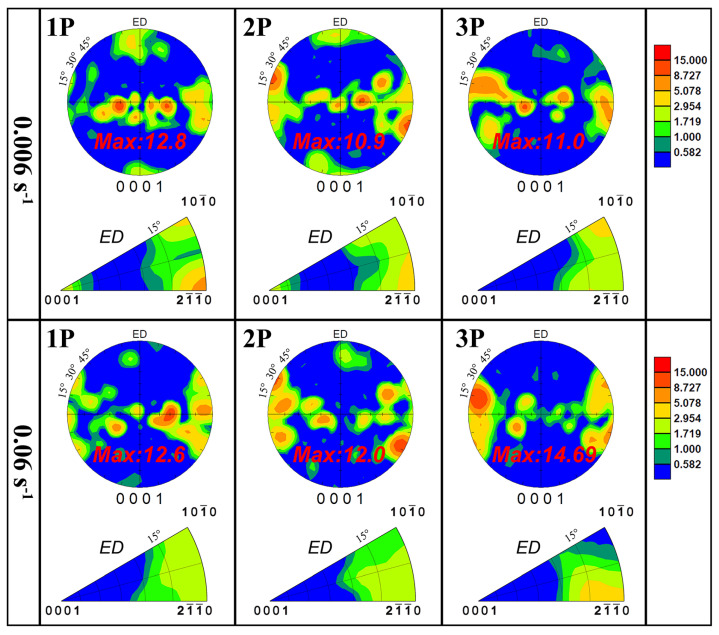
Texture characteristics of room temperature MDF samples with different parameters.

**Figure 6 materials-17-05055-f006:**
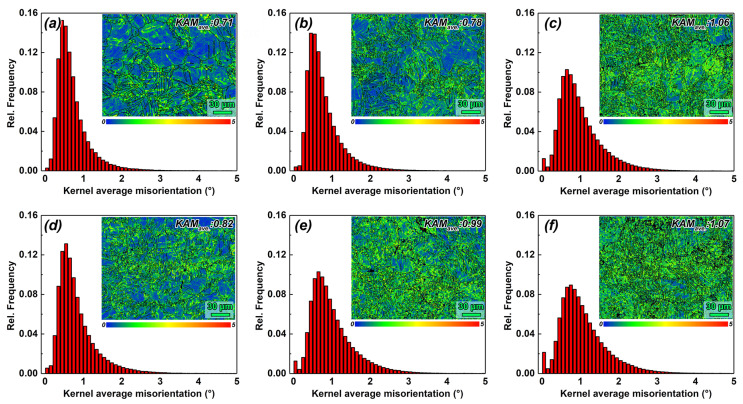
KAM maps and distribution histograms of room temperature MDF samples with different parameters. (**a**–**c**) Group 1, 0.006 s^−1^ and (**d**–**f**) Group 2, 0.06 s^−1^.

**Figure 7 materials-17-05055-f007:**
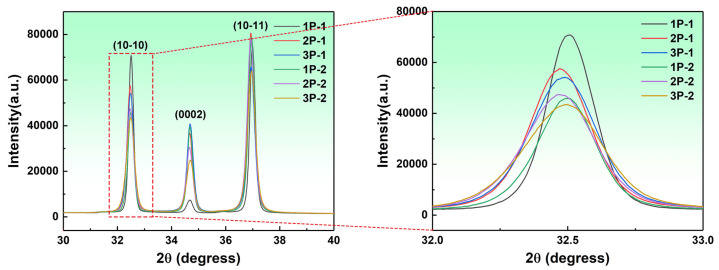
XRD patterns of room temperature MDF samples under different parameters (1: Group 1, 0.006 s^−1^, 2: Group 2, 0.06 s^−1^).

**Figure 8 materials-17-05055-f008:**
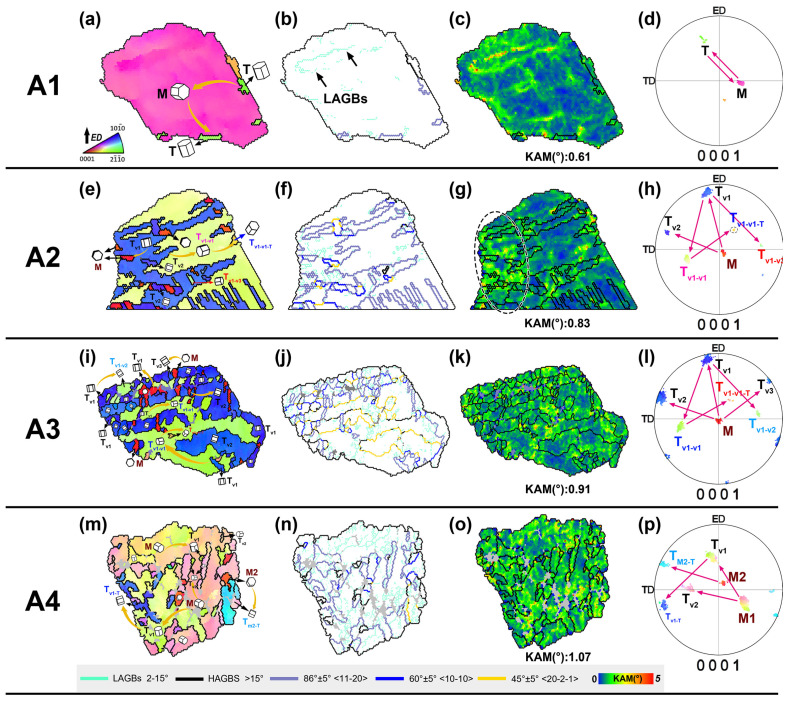
The microstructural evolution characteristics of typical grains in the A1–A4 regions of Figure 3. (**a**,**e**,**i**,**m**) IPF coloring maps, (**b**,**f**,**j**,**n**) GB maps, (**c**,**g**,**k**,**o**) KAM maps, and (**d**,**h**,**l**,**p**) discrete (0001) PFs.

**Figure 9 materials-17-05055-f009:**
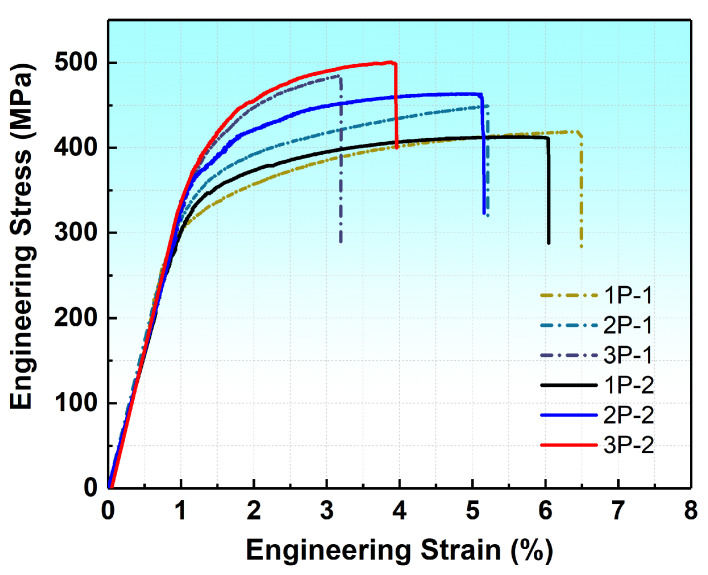
Engineering stress–strain curves of AZ80 alloys under different MDF parameters at room temperature (1: Group 1, 0.006 s^−1^, 2: Group 2, 0.06 s^−1^).

**Figure 10 materials-17-05055-f010:**
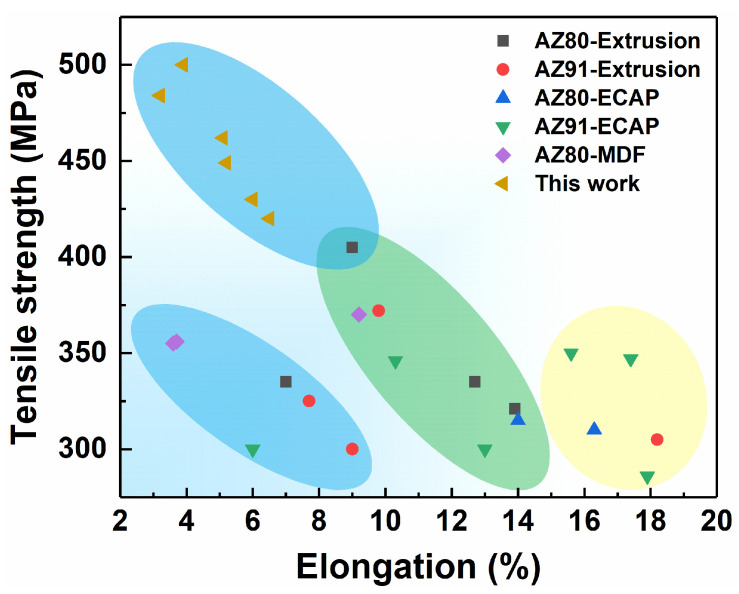
Comparison of mechanical properties of samples prepared in this study with some typical thermomechanically processed AZ80 or AZ91 Mg alloys [22,23,24,25,26].

**Figure 11 materials-17-05055-f011:**
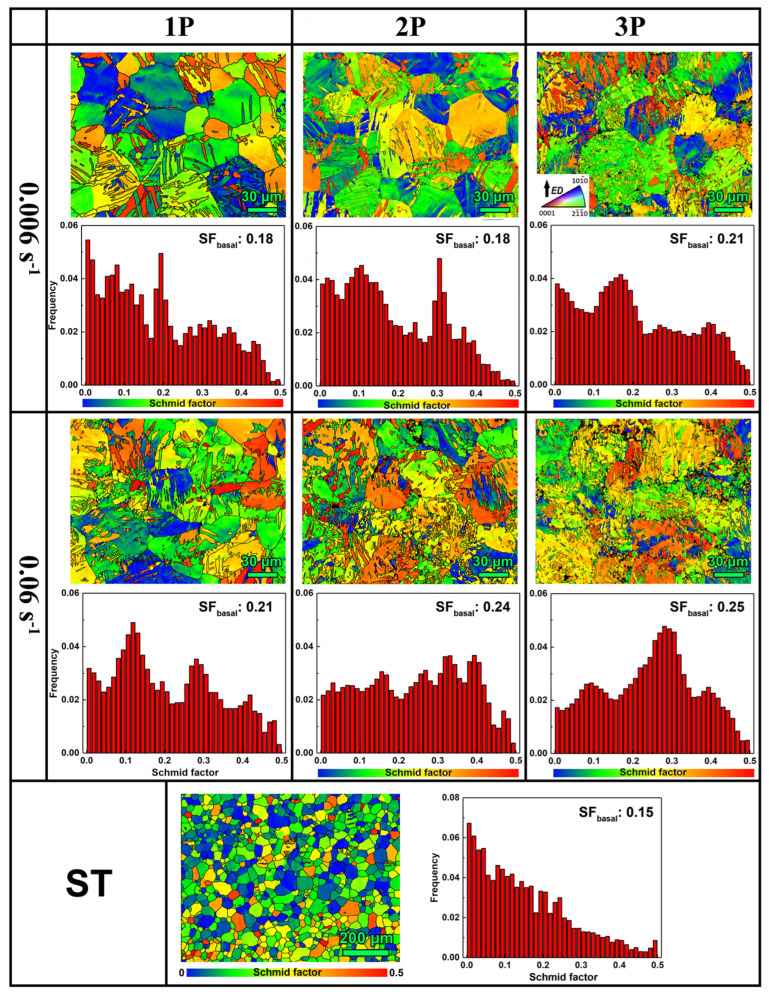
Comparison of SF for basal <a> slip when ED loading of AZ80 alloy samples after different room temperature MDF processes and ST.

**Table 1 materials-17-05055-t001:** The dislocation density of MDF samples under different deformation parameters calculated by XRD results (1: Group 1, 0.006 s^−1^, 2: Group 2, 0.06 s^−1^).

	1P-1	2P-1	3P-1	1P-2	2P-2	3P-2
Dislocation density (×10^14^/m^2^)	7.5	15.3	34.5	11.3	21.5	40.2

**Table 2 materials-17-05055-t002:** The mechanical properties of samples tested along the ED after different MDF parameters (1: Group 1, 0.006 s^−1^, 2: Group 2, 0.06 s^−1^).

	1P-1	2P-1	3P-1	1P-2	2P-2	3P-2
TYS (MPa)	285 ± 5	320 ± 6	369 ± 9	305 ± 8	345 ± 5	374 ± 8
UTS (MPa)	420 ± 8	449 ± 9	484 ± 10	430 ± 7	462 ± 7	500 ± 11
EL (%)	6.5 ± 1.1	5.2 ± 0.9	3.2 ± 1.0	6.0 ± 0.8	5.1 ± 1.1	3.9 ± 0.7

## Data Availability

Data are contained within the article.

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
