# Peer review of "Microstructural Evolution and Mechanical Properties of Extruded AZ80 Magnesium Alloy during Room Temperature Multidirectional Forging Based on Twin Deformation Mode"

_materials, 2024, doi:10.3390/ma17205055_

Round 1

Reviewer 1 Report

Comments and Suggestions for Authors

The results presented in this study are very interesting and make an important contribution to the field of high-strength magnesium alloys. However, the following improvements to the manuscript would be very helpful before final acceptance.

1) It is clearly seen that the histograms in Figure 3 are not normally distributed. Therefore, calculating the mean value does not make sense. The authors should describe such histograms more accurately.

2) The authors presented high-quality SEM images of the microstructures. However, twins are often better observed with optical microscopy. Did the authors use optical microscopy?

3) It seems that in Figure 7 there should be ‘1P-1, 2P-1, 3P-1’, not ‘1P, 2P, 3P’.

4) Lines 250 and 253. There should be superscripts.

5) Line 281. There should be ‘simultaneous’

6) Mark ‘different deformation parameters and passes’ on Fig. 8 itself or in the caption.

7) The authors write ‘the main strengthening mechanisms in room-temperature MDFed samples are grain refinement and dislocation strengthening’. This is true, but twin boundaries are also important. Moreover, a number of studies have noted that twin boundaries can simultaneously improve both strength and ductility. Discuss this point.

Author Response

Dear Reviewer:

We sincerely appreciate your review of our manuscript and apologize for any inconvenience caused during the process. Your feedback has been instrumental in improving the quality of our manuscript. Based on your and the other reviewers' comments and suggestions, we have completed the revisions. All changes can be found in the “Revised Manuscript”. The point-by-point responses are as follows:

1) It is clearly seen that the histograms in Figure 3 are not normally distributed. Therefore, calculating the mean value does not make sense. The authors should describe such histograms more accurately.

Re: The average grain size (AGS) in Figure 3 is calculated using a weighted mean. The x-axis represents the equivalent grain diameter, while the y-axis shows the percentage of area occupied by each grain size range. Therefore, this average gives a clear indication of how the grain size changes with increasing deformation passes. In other words, although the grain sizes are unevenly distributed, the average value is calculated by weighting the actual area they occupy. As a result, the average is closer to the size range with the largest area fraction, reflecting the overall trend in grain size variation.

2) The authors presented high-quality SEM images of the microstructures. However, twins are often better observed with optical microscopy. Did the authors use optical microscopy?

Re: Thank you very much for your professional advice. After undergoing room-temperature MDF, the alloy retains a high dislocation density, or residual stress, which presents significant challenges for metallographic etching. Despite numerous attempts, we were unable to effectively reveal the microstructure of the alloy under an optical microscope. Therefore, in this study, we primarily relied on EBSD for grain structure characterization. To ensure better representation of the microstructure, each sample underwent 2-3 EBSD tests. The statistical results, such as grain size and texture, were obtained by combining data from multiple measurements.

3) It seems that in Figure 7 there should be ‘1P-1, 2P-1, 3P-1’, not ‘1P, 2P, 3P’.

Re: Thank you very much for your thorough review. We have corrected this minor error.

4) Lines 250 and 253. There should be superscripts.

Re: We sincerely apologize for the oversight in creating these simple errors. We have now corrected them, and the revisions have been clearly highlighted.

5) Line 281. There should be ‘simultaneous’

Re: The spelling errors have been corrected

6) Mark ‘different deformation parameters and passes’ on Fig. 8 itself or in the caption.

Re: We apologize for the lack of clarity in the original explanation. The four representative grains in Figure 8 were selected from four regions in Figure 3. To improve clarity, we have now labeled these regions as A1-A4 in Figure 3, corresponding to the areas in Figure 8. The figure captions have also been revised accordingly. We believe these changes will eliminate any ambiguity

7) The authors write ‘the main strengthening mechanisms in room-temperature MDFed samples are grain refinement and dislocation strengthening’. This is true, but twin boundaries are also important. Moreover, a number of studies have noted that twin boundaries can simultaneously improve both strength and ductility. Discuss this point.

Re: There are some inaccuracies in the original expression. We have revised the sentence to: 'It should be considered that the main strengthening mechanisms in room-temperature MDFed samples are twinning refinement and dislocation strengthening.' In room temperature MDFed AZ80 samples, grain refinement refers to twinning-induced refinement. The formation of numerous twin boundaries significantly reduces the average grain size, producing a Hall-Petch effect that strengthens the alloy.

    There is some debate regarding the strengthening and toughening mechanisms in twinned magnesium alloys. While twin boundaries can act as obstacles to dislocation motion and strengthen the alloy, they can also serve as bridges for slip transmission. The impact of twin boundaries on alloy ductility largely depends on the loading direction, the efficiency of slip transmission across twin boundaries, and the varying micro-Hall-Petch effects. These points are discussed in the penultimate paragraph of the manuscript (lines 429-443).

Reviewer 2 Report

Comments and Suggestions for Authors

Dear authors,

the reviewed work: Microstructural Evolution and Mechanical Properties of Extruded AZ80 Magnesium Alloy during Room Temperature Multidirectional Forging Based on Twin Deformation Mode (materials-3218314) concerns an important area of science, which is the development of technology for joining materials. Magnesium alloys are currently one of the most promising construction materials with extremely low density. However, their practical use is related to increasing their strength to 500 MPa. The review article concerns the methods of mechanical processing of these alloys in the process of forging on three axes.

This article contains one fundamental flaw. It is written by a person very involved in EBSD techniques for imaging the process of material structure deformation. The article in this area is written in a kind of "jargon", which, although it makes it easier to understand its content, is only for people very involved in this technique. However, for more outsiders, it becomes difficult to read. Consider adding a few short paragraphs explaining the phenomena or explaining the symbols used, supported by the appropriate literature. It is also advisable to bring the reader closer and illustrate what, for example, the directions <a>, <c+a>, the symbols M, T, TB, TV mean.

The second noticeable shortcoming is the omission of the description of the three-axis MDC forging process. There is no good understanding of the changes that occur in the material without a clear presentation of the deformation process. Fulfilling this requirement seems to me indisputable. Referring to a literature position does not solve this problem.

Other errors or shortcomings noticed:

98 - … 30 × 30 × 30 mm3... - linear dimensions should not be recorded like this.

101-105 - Considering the initial texture… - the whole sentence is unclear and relates to the second main comment.

Fig. 1 - The diagram only talks about the assumptions of the process without explaining how it was implemented.

136 - Is AGD the same as equivalent diameter? The lack of a well-described methodology for measurements results in ambiguities. And yet, it is one of the main tests. Please complete.

157, 159 - the density of lamellae increased significantly… , … exhibited higher lamellar density… - Ok, but what do you base your statement on? There is no specific indication.

163-168 - For comparison, add a reminder of the initial grain size.

Figs. 3, 4 and next - No indication that tests were performed at room temperature (RT).

173 - The distribution of misorientation angles is denoted by the letter theta q.

172 - characteristics exhibit distinct preferential distributions… - there is no indication as to which drawings this statement applies to.

176-182 – …responds…- it would be good to mark these equivalents with coloured loops in the appropriate drawings.

184-185 - tensile twin boundaries (TBs) dominate the GB proportions - Does this mean that LAB dominate over HAB, which corresponds to TBs and GB - please write more clearly.

Fig. 5 - No explanation of what the recorded "Max" values mean.

209 – KAM - It is not obvious to the average reader what KAM is, explain it in one sentence.

316 - enhancing the Hall-Petch effect - It is not widely known what the H-P effect is - explain it briefly.

Fig. 8 - The individual diagrams do not have the designations 1P-1, etc.

355 – … typical strain hardening behaviour… - so what kind of behaviour, what does it mean?

Fig. 9 - Poor distinction of curves in b/w colours, use dashed lines.

401 - … I-type stacking faults… - What does that mean, explain.

405 - … multi-level twin activation… - What does that mean, explain.

410 -  proliferation - A rarely used concept may cause a lack of understanding of the text.

457-458 - Incorrect construction of this conclusion. First about the deformation in RT and only then about HT.

Author Response

Response to Reviewer 2:

Dear Reviewer:

Thank you very much for your thorough and patient review of our manuscript. We sincerely apologize for any inconvenience caused by the simple errors in our submission. Your comments and suggestions have been extremely helpful in improving the quality of our manuscript. We have made careful revisions, and we hope that the changes will be satisfactory. All modifications have been clearly highlighted in the revised manuscript. Our point-by-point responses are as follows:

1.This article contains one fundamental flaw. It is written by a person very involved in EBSD techniques for imaging the process of material structure deformation. The article in this area is written in a kind of "jargon", which, although it makes it easier to understand its content, is only for people very involved in this technique. However, for more outsiders, it becomes difficult to read. Consider adding a few short paragraphs explaining the phenomena or explaining the symbols used, supported by the appropriate literature. It is also advisable to bring the reader closer and illustrate what, for example, the directions <a>, <c+a>, the symbols M, T, TB, TV mean.

Re: In the manuscript, there are two types of <a>-type slip: one is basal <a>, denoted by {0001}<11-20> (line 404), and the other is prismatic <a>, represented by {10-10}<11-20> (line 436 ). These are the two most common slip systems in magnesium. <c+a> refers to pyramidal <c+a>, {11-22}<11-2-3> (line 359). We have added explanations for the slip systems and their slip directions when they first appear in the manuscript.

Additionally, 'M' stands for the Mg matrix (line 285), 'T' is the abbreviation for twin (line 286), 'TB' represents twin boundary (line 192), and 'TV' refers to twin variant (line 304). The full terms along with their abbreviations have been provided the first time they are mentioned in the manuscript.

2.The second noticeable shortcoming is the omission of the description of the three-axis MDC forging process. There is no good understanding of the changes that occur in the material without a clear presentation of the deformation process. Fulfilling this requirement seems to me indisputable. Referring to a literature position does not solve this problem.

Re: MDF is a conventional plastic deformation process that enables significant plastic deformation in magnesium alloys. However, at room temperature, the degree of plastic deformation is relatively small, with a strain of about 0.18 per pass in this experiment. Therefore, only slight changes in the sample morphology are observed during forging. For magnesium alloys, room-temperature MDF largely relies on twinning to accommodate deformation. The small strain amplitude not only sufficiently activates {10-12} twinning, which has a low critical resolved shear stress, but also prevents excessive work hardening. Since twinning in magnesium alloys is influenced by texture, altering the deformation path can induce multiple twin types. Ultimately, the interaction between twins and twin variants promotes grain refinement in the alloy.

    Due to the texture of the extruded alloy, the forging path adopted in the experiment was ED-TD-RD. Given the small strain amplitude, the forging was performed as free forging (this has been added to Section 2), with the strain on each face being ~0.06. The actual sample dimensions were 32 mm (ED) x 31 mm (TD) x 30 mm (RD), allowing easy identification of each face. We had mistakenly written the dimensions as 30 mm x 30 mm x 30 mm in the Materials and Directions section, but this has now been corrected. The experimental procedure is also described in the Materials and Methods section. We believe the forging process is straightforward, and since the sample morphology did not change significantly, we did not provide a detailed description. However, to aid readers' understanding, we have revised Figure 1, and the old image has been replaced

Other errors or shortcomings noticed:

98 - … 30 × 30 × 30 mm3... - linear dimensions should not be recorded like this.

Re: It has been revised to: 32 mm (ED) x 31 mm (TD) x 30 mm (RD). Thank you very much for your patient review.

101-105 - Considering the initial texture… - the whole sentence is unclear and relates to the second main comment.

Re: The sentence has been revised to: 'Considering the basal fiber texture characteristics of the extruded bar (Figure 2), ...'. There is a general consensus that traditional magnesium alloys (non-rare earth magnesium alloys) produced through extrusion exhibit a basal fiber texture. As previously mentioned, the activation of twinning depends on texture, and this reference serves to introduce the forging path adopted in this study. A detailed description of the texture can be found in Section 3.1 of the manuscript.

Fig. 1 - The diagram only talks about the assumptions of the process without explaining how it was implemented.

Re: As previously mentioned, to enhance readers' understanding, we have revised Figure 1, and the old image has been replaced.

136 - Is AGD the same as equivalent diameter? The lack of a well-described methodology for measurements results in ambiguities. And yet, it is one of the main tests. Please complete.

Re: AGS stands for average grain size. The EBSD data were processed using OIM Analysis software (as mentioned in Section 2). In the software, the default Grain Size represents the equivalent diameter of the grains, and AGS is calculated as a weighted average. To avoid any confusion for readers, we have added a detailed description regarding the use of OIM for processing the relevant data in the second paragraph of the Materials and Methods section (line 123-129).

157, 159 -  the density of lamellae increased significantly… , … exhibited higher lamellar density… - Ok, but what do you base your statement on? There is no specific indication.

Re: Since twins typically exhibit a lamellar structure, the term 'lamellar density' refers to the density of the twins. To avoid ambiguity, we have changed 'lamellar density' to 'twin density' (line 162 ).

163-168 - For comparison, add a reminder of the initial grain size.

Re: The revisions have been completed in accordance with your suggestions (line 170)

Figs. 3, 4 and next - No indication that tests were performed at room temperature (RT).

Re: The keyword related to room-temperature deformation have been added (line 156, Figure captions 3 and 4, etc.). Additionally, we have reviewed other sections of the manuscript and made further additions.

173 - The distribution of misorientation angles is denoted by the letter theta q.

Re: The omissions have been addressed. Thank you very much for your thorough review.

172 -  characteristics exhibit distinct preferential distributions… - there is no indication as to which drawings this statement applies to.

Re: The term 'distinct preferential distributions' refers to the three orientation peak values observed at ~45°, ~60°, and ~86° in the misorientation distributions, all of which have been mentioned. To improve clarity, we have revised the sentence to: 'After MDF, the GB characteristics exhibit three distinct preferential distributions, with major peaks at ~45°, ~60°, and ~86°.'

184-185 -  tensile twin boundaries (TBs) dominate the GB proportions - Does this mean that LAB dominate over HAB, which corresponds to TBs and GB - please write more clearly.

Re: Figure 4 only accounts for the composition of HAGBs. The boundaries of tensile twins (with a misorientation close to 86°) are classified as HAGB. This statement indicates that twinning activation is the primary deformation mechanism during room-temperature deformation, with twin boundaries constituting a significant portion of HAGBs.

Fig. 5 - No explanation of what the recorded "Max" values mean.

Re: The (0001) pole figure was calculated using OIM Analysis software. To avoid any confusion for readers, we have added a detailed description regarding the use of OIM for processing the relevant data in the second paragraph of the Materials and Methods section (Line 123-129). The texture intensity was calculated using the method of multiplying random distribution (MRD), which is a commonly employed calculation approach for textured magnesium alloys.

209 – KAM - It is not obvious to the average reader what KAM is, explain it in one sentence.

Re: KAM, or Kernel Average Misorientation, is a method for characterizing local misorientation angles. It describes the local strain distribution in crystalline materials by calculating the average misorientation angle within grains. In EBSD data analysis, grain KAM is a commonly used technique that provides insights into the local strain distribution in crystalline materials, particularly in relation to the strain distribution at grain boundaries and phase boundaries following deformation. Relevant explanations have been added to the manuscript (line 218).

316 -  enhancing the Hall-Petch effect - It is not widely known what the H-P effect is - explain it briefly.

Re: The Hall-Petch effect is a hallmark theory in materials science that has been recognized for over 70 years. This theory posits that as the grain size of metals decreases, the material itself becomes stronger.

Fig. 8 - The individual diagrams do not have the designations 1P-1, etc.

Re: There were some layout issues with Figure 8. Based on the suggestions from other reviewers and your feedback, we have made adjustments to both Figures 8 and 3 to facilitate better understanding for the readers.

355 – … typical strain hardening behaviour… - so what kind of behaviour, what does it mean?

Re: We have changed 'typical' to 'normal,' reflecting the characteristic strain hardening curve. In magnesium alloys, the activation of tensile twinning during room-temperature deformation results in distinctive features of the curve. This statement aims to emphasize that, despite the sample having a multi-scale twin structure, the deformation mechanism at room temperature is predominantly governed by dislocation slip.

Fig. 9 - Poor distinction of curves in b/w colours, use dashed lines.

Re: Thank you very much for your suggestion. The image has been adjusted accordingly.

401 - … I-type stacking faults… - What does that mean, explain.

Re: It refers to the AB'C'A-type stacking fault in magnesium, which is a very common type of stacking fault. For details, please see references [28-30]

405 - … multi-level twin activation… - What does that mean, explain.

Re: We have changed 'multi-level' to 'multi-scale.' Multi-scale twinning refers to twins with different sizes.

410 -  proliferation - A rarely used concept may cause a lack of understanding of the text.

Re: 'Proliferation' has been replaced with 'activation.'

457-458 - Incorrect construction of this conclusion. First about the deformation in RT and only then about HT.

Re: This sentence means that, unlike traditional hot deformation where dynamic recrystallization refines the microstructure of the alloy, room-temperature deformation primarily refines the grain size through twinning. To avoid ambiguity, we have made appropriate adjustments to Conclusion 1. Thank you very much for your suggestion.

Reviewer 3 Report

Comments and Suggestions for Authors

   The paper ” Microstructural Evolution and Mechanical Properties of Extruded AZ80 Magnesium Alloy during Room Temperature Multidirectional Forging Based on Twin Deformation Mode’’ can be published in Materials after some corrections:

1. The introduction chapter should be improved by adding several studies carried out so far, from the specialized literature in order to address the current state of research in this field as concretely as possible.

2. If there is a standard according to which the extruded Mg bars were made, it would be useful to add.

3. Figure 1 is unclear. It should be replaced with one of higher clarity.

5. Most of the figures have large dimensions that are not justified. Please make the figures smaller so that they don't take up much space but keep the size sufficient for the transmission of information.

5. The conclusions section is relatively short and it would be advisable to reformulate it more succinctly.

Otherwise, it is fine.

Author Response

Response to Reviewer 3:

Dear Reviewer:

We would like to express our deep gratitude to you for reviewing our paper and the constructive comments provided. We have carefully revised the manuscript and all changes have been marked in highlighting mode. We hope that all changes have been made in line with your suggestions.

1.The introduction chapter should be improved by adding several studies carried out so far, from the specialized literature in order to address the current state of research in this field as concretely as possible.

Re: The third paragraph of the introduction discusses the current preparation methods and mechanical properties of twinned magnesium alloys. Since this area is still in its early research stages, the studies are somewhat scattered. We have made every effort to encompass the prevailing research findings to date.

2.If there is a standard according to which the extruded Mg bars were made, it would be useful to add.

Re: The extruded AZ80 magnesium alloy bar used in the experiments were purchased from another company. We apologize for not being able to provide detailed information on the specific preparation process. However, it is confirmed that the extrusion temperature was 380 °C, with an extrusion speed of 5 mm/s. Following your suggestion, we have added the relevant descriptions in Figure 1.

3.Figure 1 is unclear. It should be replaced with one of higher clarity.

Re: In response to the requests of other reviewers, we have made comprehensive revisions to Figure 1. Thank you very much for your suggestion.

4.Most of the figures have large dimensions that are not justified. Please make the figures smaller so that they don't take up much space but keep the size sufficient for the transmission of information.

Re: Thank you very much for your suggestion. We sincerely apologize for any inconvenience you experienced during the review process. We maintained the larger size of the images to facilitate easier review by the reviewers. The final layout of the images will be corrected by the editor if the manuscript is accepted, so there is no need to worry excessively about this matter.

5.The conclusions section is relatively short and it would be advisable to reformulate it more succinctly.

Re: Based on your suggestion, we have made appropriate additions to Conclusion 1 and added new Conclusion 5.
